Uncovering the relationship and mechanisms of Tartary buckwheat (Fagopyrum tataricum) and Type II diabetes, hypertension, and hyperlipidemia using a network pharmacology approach

Lu Chao-Long 1 2
Zheng Qi 1
Shen Qi 2 3
Song Chi csong@icmm.ac.cn 2
Zhang Zhi-Ming zhzhang@sicau.edu.cn 1
1 Key Laboratory of Biology and Genetic Improvement of Maize in Southwest Region, Ministry of Agriculture, Maize Research Institute, Sichuan Agricultural University , Wenjiang , China
2 Institute of Chinese Materia Medica, China Academy of Chinese Medical Sciences , Beijing , China
3 Guizhou Rapeseed Institute, Guizhou Province of Academy of Agricultural Sciences , Guiyang , China
Rodriguez-Flores Juan
Electronic publication date: 2017 Nov 21
Publication date: 2017
Volume: 5
Electronic Location ID: e4042
Received 2017 Jul 21; Accepted 2017 Oct 25
Copyright: ©2017 Lu et al.
Copyright year: 2017
Copyright holder: Lu et al.
License: This is an open access article distributed under the terms of the Creative Commons Attribution License, which permits unrestricted use, distribution, reproduction and adaptation in any medium and for any purpose provided that it is properly attributed. For attribution, the original author(s), title, publication source (PeerJ) and either DOI or URL of the article must be cited.
License URL: https://creativecommons.org/licenses/by/4.0/

Keywords: Network pharmacology, Fagopyrum tataricum, Hyperglycemia, Hypertension, Hyperlipidemia, Type II diabetes, Tartary buckwheat

Funding: National Natural Science Foundation of China 3140011342 This work was supported by the National Natural Science Foundation of China (3140011342). There was no additional external funding received for this study.

==============================
Background

Tartary buckwheat (TB), a crop rich in protein, dietary fiber, and flavonoids, has been reported to have an effect on Type II diabetes (T2D), hypertension (HT), and hyperlipidemia (HL). However, limited information is available about the relationship between Tartary buckwheat and these three diseases. The mechanisms of how TB impacts these diseases are still unclear.

Methods

In this study, network pharmacology was used to investigate the relationship between the herb as well as the diseases and the mechanisms of how TB might impact these diseases.

Results

A total of 97 putative targets of 20 compounds found in TB were obtained. Then, an interaction network of 97 putative targets for these compounds and known therapeutic targets for the treatment of the three diseases was constructed. Based on the constructed network, 28 major nodes were identified as the key targets of TB due to their importance in network topology. The targets of ATK2, IKBKB, RAF1, CHUK, TNF, JUN, and PRKCA were mainly involved in fluid shear stress and the atherosclerosis and PI3K-Akt signaling pathways. Finally, molecular docking simulation showed that 174 pairs of chemical components and the corresponding key targets had strong binding efficiencies.

Conclusion

For the first time, a comprehensive systemic approach integrating drug target prediction, network analysis, and molecular docking simulation was developed to reveal the relationships and mechanisms between the putative targets in TB and T2D, HT, and HL.

Introduction

Tartary buckwheat (TB; Fagopyrum tataricum) is widely distributed in the temperate zones of the Northern Hemisphere in countries that include: China, Europe, North America, Korea, and Japan (Campbell, 1997; Ohsako & Ohnishi, 2002). TB is a medicinal and edible crop that is rich in carbohydrates, flavonoids, and chemical compounds, thus it can be used to prevent cardiovascular diseases and diabetes because of its high nutritive value and special effect on physiological regulation (Fabjan et al., 2003; Kreft, 2016; Lin, 1994; Wieslander, 1996). The rutin content in TB seed is approximately 100 times (0.8–1.7%) higher than that in common buckwheat (F. esculentum) (0.01%) (Fabjan et al., 2003). The earliest record of the medical function of TB in Chinese history traces back to about 2,000 years ago (Lin, 1994). However it has only been in recent years that TB, a health-beneficial crop, has attracted a large attention for its nutraceutical functions (Kreft, 2016; Prakash & Deshwal, 2013).

Type II diabetes (T2D), hypertension (HT) and hyperlipidemia (HL) are three major diseases with a high incidence in modern society, which have seriously damaged human health. TB has been reported to have the ability to decrease the risk of type 2 diabetes mellitus (T2DM) (Lee et al., 2012; Zhang et al., 2012); research on TB has indicated that dietary Tartary buckwheat intake attenuates insulin resistance and improves lipid profiles in patients with T2D (Qiu et al., 2016). A diet that includes TB can also reduce the blood sugar levels of patients with T2D, demonstrating that TB can contribute to the effective control of T2D (Lee et al., 2016; Zhou et al., 2015). Moreover, TB is able to antagonize the increase of capillary fragility associated with hypertension in humans (Im, Huff & Hsieh, 2003; Kreft, Knapp & Kreft, 1999). Ethanol extract from buckwheat, rutin, and quercetin have been proven to boost Akt phosphorylation and interrupt PPARγ degradation in the hepatocyte cell line, leading to improved glucose uptake (Lee et al., 2012). TB rutin-free extracts likely mediate the NO/cGMP pathways, thereby exerting endothelium-dependent vasorelaxation action (Ushida et al., 2008). The endogenous vasodilators bradykinin and NO were upregulated by TB sprouts, and, together with a lower level of the vasoconstrictor endothelin-1, relieve hypertension and oxidative stress in vivo (Merendino et al., 2014). In addition, Tartary buckwheat shell extract (TBSE) resists hyperlipidemia (Tong et al., 2006). Based on an assay used in rats fed a high-fat diet, apparent reductions in weight gain, plasma lipid concentrations, and atherogenic index were found in those rats with diets supplemented with buckwheat leaf and flowers compared with those that received no supplementation, demonstrating that buckwheat products are potential prevention and curing agents of hyperlipidemia (Brenesel et al., 2013). Although TB has been well-practiced in clinical medicine, the fundamental mechanisms and relationships between TB compounds and the interaction of these three diseases remain elusive.

TB has been demonstrated that insulin resistance was attenuated and that lipid profiles was also ameliorated in patients with T2D after dieting TB (>110 g/d) for four weeks (Qiu et al., 2016). A research on 75% ethanol extract of TB (EETB) and rutin demonstrated that both EETB and rutin suppresses the formation of fructosamine and α-dicarbonyl compounds to lower the level of AGEs (advanced glycation end-products). Hence, EETB can be considered as a potential protective agent for diabetic patients (Lee, Lee & Lai, 2015).

In the past few years there has been a considerable interest in the improvement of diabetic control to alter the glycemic impact on carbohydrates intake. A low glycemic index (GI) diet has been related to advantages in the prevention and treatment of diabetes (Ajala, English & Pinkney, 2013). Skrabanja et al. demonstrated the benefit of TB to reduce the plasma glucose. When 10 healthy testers consumed different single dose diets, boiled TB groats, bread with 50% TB groats and white wheat bread, the postpandrial plasma glucose and insulin level were tested. The results showed those who consumed TB products or groats had lower plasma glucose and insulin level compared with those who have white wheat bread (Skrabanja et al., 2001). Lan et al. reached the same conclusion from a different aspect. 10 T2D patients were randomly selected to consume TB or white wheat bread, and the postpandrial 2 h plasma glucose in the subjects having TB showed a decrease of 51% (p < 0.05) compared that in those have white wheat bread (Lan et al., 2013). Thus, TB will be a potential treatment to reduce the risk for T2D and HT.

With the development of system biology, network biology, and polypharmacology came the concept of network pharmacology, which was first proposed by Hopkins (2007) and is based on the application of multiomics and systemic biological technology. Its aim is to discover the synergistic effects and potential mechanisms of interaction between multi-components and targets by analyzing complex and multilevel interactive networks. Network pharmacology is widely used in drug discovery, target prediction, and mechanism research, especially in traditional herbal medicine (Li et al., 2015; Zhang et al., 2016). This article applies network pharmacology to investigate the mechanism of TB and its interaction with T2D, hypertension, and hyperlipidemia at the target level. Our study provides a comprehensive view of the relationships and mechanisms between TB and T2D, HT, and HL.

Materials and Methods

Composite compounds of Tartary buckwheat

We collected the composite compound data of Tartary buckwheat from the Universal Natural Products Database (UNPD) (Gu et al., 2013) (http://pkuxxj.pku.edu.cn/UNPD/, updated April 25 2013), which was specifically designed to store natural product structures for drug discovery and network pharmacology. In total, the structural information of 20 Fagopyrum tataricum compounds was collected. Detailed information on the composite compounds of Tartary buckwheat is provided in Table S1.

Known therapeutic targets of diseases

The known therapeutic target data for the treatment of T2D, HT, and HL were collected from two resources: DrugBank (Law, 2014) (http://www.drugbank.ca/, version 4.0) and Online Mendelian Inheritance in Man (OMIM) (Hamosh et al., 2000) (http://www.omim.org/, last accessed: October 31, 2015). In the DrugBank database, the targets were collected based on the following criteria: (1) they were FDA-approved therapeutic targets of the three diseases; and (2) the targets of drugs were human genes/proteins. In the OMIM database, we used the keywords “Type 2 diabetes,” “hypertension,” and “hyperlipidemia” as the queries to search known therapeutic targets of diseases. After removing the redundant results, there were 59, 279, and 20 known therapeutic targets for the treatment of T2D, HT, and HL, respectively. Detailed information on the therapeutic targets of the diseases is provided in Tables S2–S4.

Protein–protein interaction (PPI) data

PPI data were retrieved from eight public available databases: Biological General Repository for Interaction Datasets (BioGRID) (Stark et al., 2011), Human Annotated and Predicted Protein Interaction Database (HAPPI) (Chen, 2009), Human Protein Reference Database (HPRD) (Keshava Prasad et al., 2009), High-quality INTeractomes (HINT) (Jishnu & Yu, 2012), Molecular INTeraction Database (MINT) (Chatraryamontri, 2010), Online Predicted Human Interaction Database (OPHID) (Brown & Jurisica, 2005), Database of Interacting Proteins (DIP) (Xenarios et al., 2002), and Search Tool for the Retrieval of Interacting Genes/Proteins (STRING) (Szklarczyk et al., 2011). Detailed information from the eight databases is provided in Table S5.

Target prediction of composite compounds

The Bioinformatics Analysis Tool for Molecular mechANism of Traditional Chinese Medicine (BATMAN-TCM) database (Liu et al., 2016b), which is aimed at the target prediction of composite compounds of tartar buckwheat, was employed. In this database, there are 6 basal principles for the measurement of drug-drug similarity that are based on chemical structure (including FP2 fingerprint-based and functional group-based similarity scores), side effects, the Anatomical, Therapeutic and Chemical (ATC) classification system, drug-induced gene expression, and the text mining score of chemical-chemical associations, and 3 scores to measure protein-protein similarity respectively based on protein sequence, closeness in a protein interaction network and Gene Ontology (GO) functional annotation. The default parameters were set for the putative targets of composite compounds of Tartary buckwheat.

Network construction and analysis

The TB-composite compound-putative target-known therapeutic target network was constructed to find the key target. Then, the target-pathway network was established to find the relationship between the pathways and the key targets. The key target-pathway networks would be used to explore core pathways that could play an important role in the interaction mechanism of TB and the three diseases.

Cytoscape (Shannon et al., 2003) (http://www.cytoscape.org/, version 3.2.1) and NAViGaTOR (http://ophid.utoronto.ca/navigator/, version 2.3) were employed to directly visualize the networks. In addition, four topological features (‘Degree,’ ‘Betweenness,’ ‘Closeness,’ and ‘K core’) were calculated using the igraph package, which is a powerful tool for topological graphing in R (https://cran.r-project.org/).

The networks were simplified using the following procedure: (A) we deleted the nodes that had degree values of less than 2-fold the median of all of the nodes in the network,and we then used the retained nodes to construct the hub network. (B) We retained the nodes that were greater than the corresponding median values of the four topological features: ‘Degree,’ ‘Betweenness,’ ‘Closeness,’ and ‘K core’ (Li et al., 2007).

Pathway enrichment analyses

The clusterProfiler package of R software (Yu et al., 2012) was employed to classify the biological terms and to analyze the gene cluster enrichment automatically. The latest data were obtained from the Kyoto Encyclopedia of Genes and Genomes (KEGG) database (Kanehisa & Goto, 1999) for KEGG pathway enrichment analyses. P-values were set at 0.05 as the cut-off criterion, and the results of both analyses were annotated by Pathview (Luo, 2013) in the R Bioconductor package (https://www.bioconductor.org/).

Molecular docking simulation

LibDock was implemented in the Discovery Studio 2.5 (DS 2.5) software to determine the molecular docking simulation. It is an efficient and powerful tool to validate the binding ability of candidate targets to composite compounds of herbs. All of the crystal structure data of the targets were directly retrieved from the RCSB Protein Data Bank (http://www.rcsb.org/pdb/home/home.do, last accessed Dec 27, 2016). The high-resolution crystal structure was a priority for verification. We then utilized the customizable scoring function from LibDock to calculate the docking score to measure the binding ability of each candidate target of the corresponding compound. The docking scores of the candidate targets with a strong binding ability to their corresponding compounds were greater than the median value of the all of the docking scores.

Results and Discussion

Putative targets for Tartary buckwheat

A total of 20 ingredients in TB were retrieved from the Universal Natural Products Database (UNPD). The detailed information about these molecules is provided in Table S1. Following the drug target predicted by BATMAN-TCM, 97 putative targets of the 20 ingredients of TB were identified (Table S6). In addition, known therapeutic targets of the three diseases were collected from two public databases (described in the ‘Materials and Methods’ section). We obtained 59, 279, and 20 known therapeutic targets for the treatment of T2D, HT, and HL, respectively. Interestingly, 8 and 1 putative targets of TB were significant proteins for HT and HL, respectively (Fig. 1). PPARG (Peroxisome proliferator-activated receptor gamma) was shared by PT, T2D, and HT; ABCA1 (ATP-binding cassette sub-family A member 1) was shared by PT, T2D, and HL; PPARA (Peroxisome proliferator-activated receptor alpha) was shared by PT, HT, and HL; and SLC6A4 (Sodium-dependent serotonin transporter) was shared by PT, T2D, HT, and HL (Table S7). SLC6A4 plays a significant role in regulating serotonin for the availability of other serotonin system receptors (Comings et al., 1999; Zhang et al., 2007). PPARA is a key regulator of lipid metabolism (Gorla-Bajszczak et al., 1999; Laurent et al., 2013). ABCA1 functions as a key gatekeeper influencing intracellular cholesterol transport (Kathiresan et al., 2008; Singaraja et al., 2003). PPARG is important for its regulation of adipocyte differentiation and retention of glucose homeostasis (Katanotoki et al., 2013; Mukherjee et al., 1997; Park et al., 2011). Out of the 97 putative targets of TB compounds, there were 13 that were related to these three diseases, suggesting the possibility of TB as their treatment.

Figure 1 Venn diagram showing the overlap of significant targets in PT, T2D, HT, and HL.

PT, putative targets, green; T2D, type II diabetes, blue; HT, hypertension, pink; and HL, hyperlipidemia, yellow.

Identification of the underlying pharmacological mechanisms of TB on the three diseases

A network was constructed based on TB-composite compound-putative targets and known therapeutic targets of the diseases to elucidate the pharmacological mechanisms of TB on these three diseases. Protein-protein interaction (PPI) data of the putative targets and the known therapeutic targets of the three diseases were collected from eight public PPI databases (as described in the Materials and methods section). The network consisted of 455 nodes and 1,748 edges in total. Two-fold the median value of degree was set as the threshold. The network was reconstructed after deleting the nodes that were less than the threshold. As a result, the nodes were reduced from 455 to 132, and the edges from 1,748 to 1,010 in the reconstructed network. In order to determine the key targets in the network, four attributes (‘Degree,’ ‘Betweenness,’ ‘Closeness,’ and ‘K core’) were calculated in the topological networks. The network was further simplified with these four values, and the key target information was finally obtained. The four topological features were used to retain the nodes that were over the median in the rebuilt network. The median values of ‘Degree,’ ‘Betweenness,’ ‘Closeness,’ and ‘K core’ were 10.0000, 38.2517, 0.0034, and 8.0000, respectively. Therefore, targets with ‘Degree’ > 10.0000, ‘Betweenness’ > 38.2517, ‘Closeness’ > 0.0034, and ‘K core’ > 8.0000 were defined as the key targets (Table S8). As a result, the network that was rebuilt with the key targets had 29 nodes and 163 edges (Fig. 2).

Figure 2 Interaction network between chemical components of TB, their putative targets, and known therapeutic targets of the three diseases built and visualized with Cytoscape.

Blue line, linked PT and their targets; purple, linked T2D and their targets; green, linked HL and their targets; yellow, linked HT and their targets; and light blue, linked chemical components and their targets.

Lines with different colors were employed to show their importance from the targets to their corresponding sources (PT, T2D, HT, and HL) in our network, and diameter was used to denote degree. A larger node diameter represented a higher degree in the network, and vice versa. Similarly, with the targets, those with the higher degree played a more important role in the network. Compared with all of the other targets, SRC (Proto-oncogene tyrosine-protein kinase Src), JUN, and IL1B (Interleukin-1 beta) had the highest degree number (19), which indicated that these targets play key roles in the regulation of T2D, HT, and HL. Our results agreed well with previous research, demonstrating that JUN modulated smooth muscle cell proliferation in response to vascular angioplasty (Hu et al., 1997), SRC modulated endothelial cell angiogenic activities (Desjarlais et al., 2017), and that IL1B has been associated with the development of chronic inflammation in obesity(Maldonado-Ruiz et al., 2017; Osborn et al., 2008). Moreover, PT, T2D, HL, and HT were linked to 10, 4, 1, and 17 key targets in the network, respectively. Specifically, UNPD28717 (salicylic acid) was linked to nine key targets, indicating that it may mediate these targets to regulate blood-vessel dilation, inflammatory cytokine, and adipose tissue (Liu et al., 2016a; Tang & Dong, 2017).

Pathway analysis to explore the underlying mechanisms of TB and the three diseases

In order to investigate the relationship and mechanisms between the targets and the pathways, a target-pathway network was constructed (as described in the ‘Materials and Methods’ section). The KEGG database was used to describe KEGG pathways, to systematically analyze gene functions, and to provide a reference knowledge base linking genomes to functional information. In total, 48 pathways were obtained by igraph to analyze the KEGG enrichment of key targets. The pathway-target network contained 76 nodes (48 pathways and 28 targets) and 352 edges. The median values of ‘Degree,’ ‘Betweenness,’ ‘Closeness,’ and ‘K core’ were 7.0000, 23.3158, 0.0059, and 6.0000, respectively (Table S9). AKT2, IKBKB, RAF1, TNF, and CHUK were in the top-ranking positions in the pathway-target network. Additionally, the results indicated that some targets had been hit by multiple pathways in the pathway-target network. ATK2, IKBKB, RAF1, CHUK, TNF, JUN, and PRKCA were linked by 42, 32, 29, 26, 26, 24, and 17 pathways (Fig. 3). AKT2 (RAC-beta serine/threonine-protein kinase) is responsible for the regulation of glucose uptake by mediating insulin-induced translocation (Hers, Vincent & Tavaré, 2011; Zhang et al., 2006). IKBKB (inhibitor of nuclear factor kappa-B kinase subunit beta) plays an essential role in the NF-kappa-B signaling pathway (hsa04064), which is activated by multiple stimuli, such as inflammatory cytokines (Mercurio et al., 1997). RAF1 (RAF proto-oncogene serine/threonine-protein kinase) is involved in proliferation and angiogenesis (Chong, Lee & Guan, 2001; Yao et al., 1995).

Figure 3 The target-pathway network.

Pink dots are targets, purple diamonds are pathways, and the dot size and diamond size represent node degree value.

Figure 4 The target-composite compound network.

Blue dots are chemicals, while pink dots are key targets. The size of the edges are docking scores, and the size of the dots are node degrees.

Pathways related to these targets were shown to have more significant features (Fig. 3). Among the pathways, hsa5200 (a pathway in cancer) exhibited the highest number of target connections (degree = 15), followed by hsa05418 (a fluid shear stress and atherosclerosis pathway) with 14 targets, and hsa04151 (a PI3K-Akt signaling pathway) with 11 targets, respectively. These high-degree pathways were closely related to vascular conditions and inflammation. The hsa5200 pathway was the underlying mechanism of inflammation and involved in multiple targets, such as PPARG, JUN, CHUK, IKBKB, AKT2, and RAF1 (Andersen et al., 2017; Kolb, Sutterwala & Zhang, 2016). The fluid shear stress and atherosclerosis pathway plays an important role in atherosclerosis, and it is associated with systemic risk factors, including hypertension, hyperlipidemia, and diabetes mellitus (Cheng et al., 2006; Malek, Alper & Izumo, 1999). The PI3K-Akt signaling pathway is one of the best characterized downstream effectors of insulin and belongs to insulin-activated intracellular pathways (Westermeier et al., 2011). In addition, we found that some pathways discovered in this study, such as the insulin resistance pathway, AGE-RAGE (Advanced glycation end products) signaling pathway (Hegab et al., 2012; Roy, 2013), and insulin signaling pathway(Table S10), have a direct relationship with the three diseases. Overall, the key targets are significantly associated with these pathways that might play a role in the progression of the three diseases.

Molecular docking validation

The computational docking technique, as a structure-based method, is an invaluable tool in drug discovery and design. This technique can help researchers discover the relationships between the constituents of TCM and network targets (Luo et al., 2014a; Luo et al., 2014b; Yu et al., 2016). The Libdock module of the DS2.5 software was used for molecular analog docking to obtain the effective dockings of TB and its key targets and to get docking scores. The score was greater than the median value (86), indicating a strong binding capacity between the composite components of TB and the molecular targets in this study. In total, we obtained 174 docking results. Among these results, JUN received the highest score of 170.967 with chemical UNPD51223 (Table S11). The docking score results were used to construct the target-composite component network (Fig. 4). The target-composite compound network contains 38 nodes (21 targets and 17 composite compounds) and 174 edges. In addition, the line width shows the docking value, meaning a thicker line represents a higher docking value and vice versa. As a result, JUN, TNF, PPARA, PPP2CA, PPARG, and IKBKB had a high degree value and larger molecular analog docking scores (Table S12). These targets were proven to bind to multiple chemicals.

Conclusions

Tartary buckwheat has a very high nutritional value and is of great medicinal value to treat T2D, hypertension, and hyperlipidemia. Our studies investigated the relationships between TB and the three diseases using network pharmacology. In total, 97 putative targets were obtained from 20 composite components of TB. The TB-composite compound-putative target-known therapeutic target networks reveals that 28 key targets play a significant role in their interplay. To further study the relationships and underlying mechanisms between the key targets and pathways, key target-pathway networks were constructed. ATK2, IKBKB, RAF1, CHUK, TNF, JUN, and PRKCA were mainly involved in fluid shear stress and the atherosclerosis pathways, pathways in cancer, and the PI3K-Akt signaling pathway. Moreover, 174 candidate molecular analog docking results were obtained based on the calculation of chemical molecules from the molecular analog docking experiment. These results provide strong evidence that TB is a potential treatment to T2D, HL and HT, and that this comprehensive systemic approach integrating drug target prediction, network analysis, and molecular docking simulation is a useful tool to reveal relationships and mechanisms between the putative targets in TB and T2D, HT, and HL.

Supplemental Information

Supplemental Information 1 Information about the composite compounds of Fagopyrum tataricum

Click here for additional data file.

Supplemental Information 2 Supplementary Tables 2, 3, and 4

Supplementary Table S2: Known therapeutic targets for Type II diabetes.

Supplementary Table S3: Known therapeutic targets for hypertension.

Supplementary Table S4: Known therapeutic targets for hyperlipidemia.

Click here for additional data file.

Supplemental Information 3 Supplementary Table 5: Detailed information on eight existing protein-protein interaction databases

Click here for additional data file.

Supplemental Information 4 Supplementary Table 6: Prediction of putative targets for composite compounds of Fagopyrum tataricum

Click here for additional data file.

Supplemental Information 5 Overlap of significant targets in PT, T2D, HT, and HL

Click here for additional data file.

Supplemental Information 6 Four topological feature values of key targets in the TB-composite compound-putative target-known therapeutic target network

Click here for additional data file.

Supplemental Information 7 Four topological feature values of key targets in the target-pathway network

Click here for additional data file.

Supplemental Information 8 Detailed information on the pathway of key targets

Click here for additional data file.

Supplemental Information 9 Detailed docking information on key targets and composite compounds

Click here for additional data file.

Supplemental Information 10 Four topological feature values of targets in the target-composite compound network

Click here for additional data file.

Additional Information and Declarations

Competing Interests

Author Contributions

Data Availability

The authors declare there are no competing interests.

Chao-Long Lu conceived and designed the experiments, performed the experiments, analyzed the data, contributed reagents/materials/analysis tools, wrote the paper, prepared figures and/or tables.

Qi Zheng performed the experiments, analyzed the data, wrote the paper, prepared figures and/or tables.

Qi Shen performed the experiments, contributed reagents/materials/analysis tools, prepared figures and/or tables.

Chi Song and Zhi-Ming Zhang conceived and designed the experiments, reviewed drafts of the paper.

The following information was supplied regarding data availability:

The raw data can be found in the Supplemental Files.

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
