# Peer review of "Uncovering the relationship and mechanisms of Tartary buckwheat (Fagopyrum tataricum) and Type II diabetes, hypertension, and hyperlipidemia using a network pharmacology approach"

_PeerJ, doi:10.7717/peerj.4042_

## Round 0.1 · original submission · Minor Revisions

Please review the comments from the reviewers and make their requested edits to the manuscript.

Reviewer 1 ·

Basic reporting

Nice and highly relevant paper. It is a report on important results of the investigation, based on a comprehensive systemic approach integrating drug target prediction, network analysis, and molecular docking simulation, developed to reveal the relationships and mechanisms between the putative targets in Tartary buckwheat and Type II diabetes, hypertension, and hyperlipidemia.

Authors are reporting that some pathways discovered in the study, such as the insulin resistance pathway, AGE-RAGE (Advanced glycation end products) signaling pathway, and insulin signaling pathway, have a direct relationship with the three studied diseases.

They revealed the relationship of Tartary buckwheat and the three diseases to provide more information on the clinical applications of Tartary buckwheat and to further research on Tartary buckwheat properties and quality.

Mainly clear professional English is used. Paper with nice novel results. Title of the manuscript is suitable, abstract is enough informative. Literature references are sufficient. Conclusions are of broad interest and based on the results. Abstract, Introduction, Results and Discussion are clearly presented. Professional structure of manuscript is well performed. Figures are clear, but the reviewer has a specific suggestion to Fig. 1.
Results are self-contained with the results relevant to hypotheses.

Experimental design

Research questions are well defined and within the aims and the scope of the journal. Materials are accordingly defined. Methods are suitable, properly described and used, in a way that is possible to replicate experiments. The investigation is performed to high technical standards. It is no ethical problem involved.

Validity of the findings

Data are obtained in a sound way, they are statistically evaluated in a proper way.
Conclusions are very well stated and based on the results. Discussion and conclusions are sound and relevant.

Additional comments

Specific suggestions:

Line 1 and elsewhere in the manuscript: Instead of »tartary buckwheat« correct to »Tartary buckwheat«, where applicable. In difference to »common buckwheat« (with lower-case first letter), »Tartary buckwheat« should be in English always written by upper-case first letter; this way of writing was internationally accepted, among others in the discussion on International Buckwheat Research Association (IBRA) Assembly in 2013, as word »Tartary« comes from the name of the Tartar people.

Line 51, delete: amount of

Line 67, instead of: shell, better: husk

Figure 1, Nice and illustrative figure, however »PT« letters should be better placed outside the »T2D« area, to be more clear, including to be clear on black/white prints.

Reviewer 2 ·

Basic reporting

The manuscript reported new and relevant results on systemic approach integrating drug target prediction with the network analysis, and molecular docking simulation. Results are important for understanding relationships and mechanisms between the putative targets in Tartary buckwheat and Type II diabetes, hypertension, and hyperlipidemia.

Professional English of the manuscript should be improved

Experimental design

Experimental design is well performed and within the aims and the scope of the journal. Materials and methods are well defined and used. It is thus possible to replicate experiments. The investigation is performed according to high technical standards. There seems that no ethical issues are involved.

Validity of the findings

Data are obtained in a suitable way,and correctly evaluated.
Conclusions are properly based on the experimental results.

Additional comments

Reviewer suggests improvement of English style.

In the title and manuscript Tartary buckwheat should be written with the capital first letter.

Text in lines 44-47 is not clear enough, needs rewriting.
Line 54: correct »which have«
Line 54, punctuations are not correct.
Line 132. punctuations are not correct.
Line 263, the reference should be inserted.

Conclusions are not clear enough, they should more clearly pointed out the purpose, results and focus of the paper.

---

## Round 0.2 · Minor Revisions

The revised manuscript is considerably improved, and additional reviews are not needed. The value of this publication is high, and I hope this article will be published soon. However, I have three small but important issue with the conclusions that should be addressed.

1. The article proposes Tartary buckwheat as a potential treatment for T2D, which is a bit over-zealous. Could it actually be used as a treatment for individuals with T2D? Is there evidence in the literature that you can cite with regards to treatment and cure of T2D using diet modification alone? I would suggest additional discussion of the literature regarding this subject, and softening of the language to proposing the potential for TB consumption for the REDUCTION of T2D risk in individuals who have not yet developed T2D, as opposed to a TREATMENT for individuals diagnosed with TB.

2. TB is a food with high fiber and a low glycemic index, both factors that are known to reduce risk for T2D and HT. Please mention this aspect of TB in the discussion, and cite relevant literature.

3. The potential benefits of TB consumption should be measured in comparison to consumption of other foods, such as rice or wheat. The value of the manuscript would be greatly enhanced by conducting the same analysis on a different grain, such as rice, wheat, oat, or quinoa. The authors should demonstrate the EXCESS value of TB compared to other grains.

---

## Round 0.3 · accepted · Accept

The manuscript takes a novel and valuable approach at evaluating the medicinal value of food, which impressed myself and the reviewers.